# Physicochemical Stability of Generic Thiotepa Concentrate and Ready-to-Administer Infusion Solutions for Conditioning Treatment

**DOI:** 10.3390/pharmaceutics15020309

**Published:** 2023-01-17

**Authors:** Helen Linxweiler, Judith Thiesen, Irene Krämer

**Affiliations:** Department of Pharmacy, University Medical Centre of Johannes Gutenberg-University, Langenbeckstraße 1, 55131 Mainz, Germany

**Keywords:** thiotepa, physicochemical stability, concentrate, parenteral preparation, ready-to-administer, conditioning treatment

## Abstract

The objective of this study was to determine the physicochemical in-use stability of recently approved Thiotepa Riemser concentrate in the original vial and diluted ready-to-administer (RTA) infusion solutions in prefilled glucose 5% and 0.9% NaCl polyolefin bags. Thiotepa Riemser 10 mg/mL concentrates and infusion solutions (1 mg/mL, 2 mg/mL, 3 mg/mL) were prepared in triplicate and stored at 2–8 °C or 25 °C for 14 days. Thiotepa concentrations were determined using a stability-indicating RP-HPLC assay. In parallel, pH and osmolality were measured. Sub-visible particles were counted on day 0 and 14. Thiotepa Riemser concentrate was revealed to be stable for 14 days when stored at 2–8 °C, or for 24 h when stored at 25 °C. Thiotepa concentrations in infusion solutions stored at 2–8 °C remained above 95% of the initial concentrations for at least 14 days, regardless of the type of vehicle solution. When stored at 25 °C, thiotepa infusion solutions in glucose 5% proved to be physicochemically stable for 3 days (1 mg/mL), 5 days (2 mg/mL) or 7 days (3 mg/mL). Thiotepa infusion solutions in 0.9% NaCl remained physicochemically stable for 5 days (1 mg/mL) or 7 days (2 mg/mL, 3 mg/mL). At these points in time, the specification limit of ≤0.6% monochloro-adduct was fulfilled. In parallel, an elevation of the pH values was registered. Thiotepa concentrates and infusion solutions should be stored at 2–8 °C due to temperature-dependent physicochemical stability, and for microbiological reasons. Glucose 5% infusion solution is recommended as a diluent, and stability-improving nominal 2 mg/mL to 3 mg/mL thiotepa concentrations should be obtained.

## 1. Introduction

Thiotepa (1,1′,1″-phosphinothioylidynetrisaziridine; for the chemical structure see Figure 1) is a trifunctional alkylating agent chemically and pharmacologically related to nitrogen mustard. Its antitumour activity is caused by interaction with the DNA, mainly by the formation of DNA crosslinks and the intracellular release of aziridine, causing monofunctional alkylation of nucleic acids [1]. Thiotepa has been in clinical use for more than 50 years for the treatment of various solid tumours, and is today of special interest in high-dose regimens followed by blood stem cell transplantation. In 2021, generic Thiotepa Riemser (Esteve Pharmaceuticals GmbH, Berlin, Germany) was approved by the European Medicines Agency (EMA). The licensed indications are conditioning treatment prior to haematopoietic progenitor cell transplantation in haematological diseases and solid tumours in adult and paediatric patients. The recommended doses range from 120 mg/m^2^/day to 481 mg/m^2^/day, dependent on the indication [2].

Thiotepa Riemser is marketed as a powder for a concentrate containing 15 or 100 mg thiotepa per vial. Identical to the originator Tepadina^®^ (Adienne S.r.l. S.U., Caponago, Italy) the finished product is a terminally sterilised, lyophilised powder not containing any excipients [3]. Reconstitution of Thiotepa Riemser with 1.5 or 10 mL of water for injection results in 10 mg/mL thiotepa concentrate solutions with a pH between 5.5 and 7.5 [3]. Prior to administration, the hypotonic concentrate has to be diluted with 0.9% NaCl infusion solution in order to obtain a final thiotepa concentration between 0.5 and 1 mg/mL [3]. The concentrate is physicochemically stable for 8 h when stored at 2–8 °C, whereas diluted solutions are stable for 24 h when stored at 2–8 °C and for 4 h when stored at 25 °C [2]. Thiotepa can polymerize to form insoluble polymeric derivatives. Therefore, in-line filtration of the infusion solutions next to the patient is obligatory according to the summary of product characteristics (SmPC) [2]. In the SmPC of the formerly licensed Thiotepa “Lederle” 15 mg lyophilised product alongside 0.9% NaCl infusion solution, glucose 5% (G5), Ringer solution and Ringer lactate solution were recommended as a diluent solution [4]. 

Previous in-use stability studies of various licensed thiotepa medicinal products formulated as lyophilised powder revealed that the stability of thiotepa in aqueous solutions depends on temperature, concentration, type of diluent and pH [5,6]. Thiotepa is most stable in the pH range 7–11 [7]. In aqueous solutions, thiotepa undergoes ring-opening solvolysis of the activated aziridine rings with nucleophilic reagents [7]. The ring opening reaction is accelerated in acidic media, but also occurs in basic and neutral media [8]. In the absence of chloride ions, mono-, di- and trihydroxy-adducts are formed (compare Figure 1). In the presence of chloride ions, e.g., after dilution with 0.9% NaCl solution, mono-, di- and trichloro-adducts are preferably formed, as well as partly chlorinated and hydroxylated adducts [7]. The trihydroxy- and trichloro-adducts can be considered as end-products of the degradation reactions in the absence and presence of sodium chloride, respectively (compare Figure 1). In acidic media and at elevated temperatures, thiotepa undergoes P-N cleavage to give aziridine (ethylenimine) [9]. Aziridine is also formed intracellularly by chemical or enzymatic hydrolysis of thiotepa, thereby acting as a prodrug [9].

**Figure 1 pharmaceutics-15-00309-f001:**
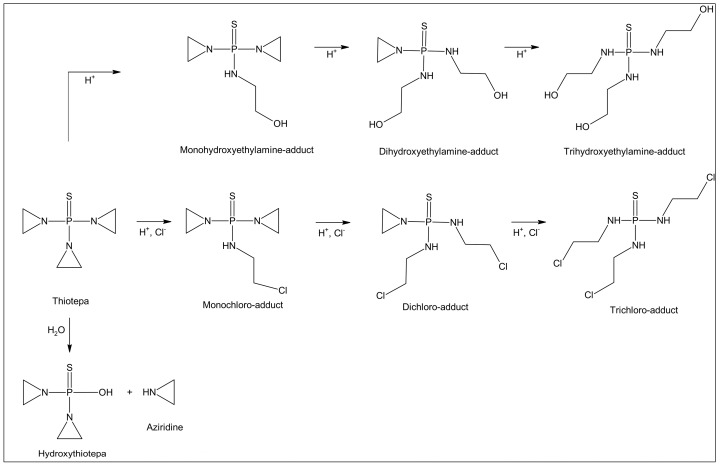
Degradation scheme of thiotepa in the absence and presence of sodium chloride, adapted from van Maanen et al. [7].

Monographs for thiotepa active substances and thiotepa injections are available in the BP and USP [10,11,12,13]. According to the thiotepa-for-injection monographs, the thiotepa content amounts to 95%–110% of the labelled amount. The threshold for unspecified impurities/secondary peaks is set at 0.2% of the thiotepa content. The content of the monochloro-adduct may not exceed 0.15%. It should be noted that higher percentage rates of the monochloro-adduct are accepted elsewhere [6]. 

The objective of our study was to determine the physicochemical in-use stability of dissolved Thiotepa Riemser lyophilised powder and diluted Thiotepa Riemser infusion solutions in common infusion fluids. The study was designed with regard to the known stability-determining factors (i) type of diluent solution and pH (0.9% NaCl, G5), (ii) thiotepa concentration (nominal concentration 1, 2, 3 mg/mL), and (iii) storage temperature (refrigerated, room temperature). The type of diluents and the concentration range were chosen because of their relevance for conditioning treatment in stem cell transplantation centres.

## 2. Materials and Methods

In this study, thiotepa solutions are determined to be stable when thiotepa concentrations are in the range of 95 and 110% of the initial concentration measured (BP, USP) [10,11]. Furthermore, the peak area of the monochloro-adduct must not exceed 0.6% of the thiotepa parent peak area [6]. Limits for the pH-values of the concentrate are set to 5.5–7.5 according to the BP and manufacturer specification [3,10]. In diluted solutions, limits of particles ≥10 µm are set to ≤6000 and particles ≥25 µm to ≤600, according to Ph. Eur., 2.9.19 and USP <788>.

### 2.1. Chemicals and Reagents

Mobile phase: water HPLC grade (AppliChem GmbH, Darmstadt, Germany, batches 1O012859, 1X012088, 1J012256, 1J011606, 1C013185), acetonitrile HPLC grade (Honeywell International, Morristown, NJ, USA, batches L236M, L0190, L1810), di-potassium hydrogen carbonate (AppliChem GmbH, Darmstadt, Germany, batch 0001912363; Carl Roth, Karlsruhe, Germany, batch 439289808), sodium hydroxide 1 M (AppliChem GmbH, Darmstadt, Germany, batch 0001889275). Thiotepa standard solution: thiotepa chemical reference substance (CRS) United States Pharmacopeia (USP) (batch R11380). Thiotepa test solutions: Thiotepa Riemser 100 mg powder for a concentrate (Esteve Pharmaceuticals GmbH, Berlin, Germany, batch 202191). Solvent and diluent for samples: water for injection 100 mL (B. Braun Melsungen AG, Melsungen, Germany, batch 213468091), 0.9% NaCl injection solution 10 mL (Fresenius Kabi, Bad Homburg, Germany, batch 20PMH013), 5% glucose injection solution 10 mL (B. Braun Melsungen AG, Melsungen, Germany, batch 21113012). Vehicle solutions: pre-filled 0.9% NaCl 100 mL freeflex^®^ polyolefin infusion bags (Fresenius Kabi, Bad Homburg, Germany, batches 13QDS092, 13QHS181), pre-filled 5% glucose 100 mL freeflex^®^ polyolefin infusion bags (Fresenius Kabi, Bad Homburg, Germany, batch 13QBS093).

### 2.2. Preparation of Test Solutions and Samples

#### 2.2.1. Thiotepa Concentrate 10 mg/mL

Test solutions of Thiotepa Riemser concentrate 10 mg/mL were prepared by dissolving the powder with 10 mL water for injection each. Triplicate test solutions were stored in the original vials either at 2–8 °C in the refrigerator or at 25 °C, 60% air humidity, 24 h daylight in a climatic chamber. Immediately after preparation and on day 1, 3, 5, 7 and 14, a sample of 1 mL concentrate was withdrawn. 0.1 mL aliquots were diluted with water HPLC grade to a nominal concentration of 1 mg/mL thiotepa and used for HPLC assays.

#### 2.2.2. Thiotepa Infusion Solutions

Diluted Thiotepa Riemser infusion solutions of the nominal concentrations 1 mg/mL, 2 mg/mL and 3 mg/mL were prepared in triplicate using prefilled 100 mL 0.9% NaCl and 100 mL G5 freeflex^®^ polyolefin bags. To reach a final volume of 100 mL infusion solution, the average overfilling of 6 mL and the volume of thiotepa concentrate to be added were withdrawn from the prefilled infusion bags (16 mL, 26 mL, 36 mL). Afterwards, 10 mL, 20 mL, or 30 mL of freshly prepared Thiotepa Riemser concentrates were added to the infusion bags. Test solutions were either stored at 2–8 °C in the refrigerator or at 25 °C, 60% air humidity, 24 h daylight in a climatic chamber. Immediately after preparation and on day 1, 3, 5, 7 and 14, samples of 1 mL (applies to 2 mg/mL and 3 mg/mL thiotepa infusion solutions) or 2 mL (applies to 1 mg/mL thiotepa infusion solutions) were withdrawn from each infusion bag. For the HPLC assay, 0.5 mL aliquots (2 mg/mL samples) or 0.33 mL aliquots (3 mg/mL samples) were diluted to a concentration of 1 mg/mL with G5 or 0.9% NaCl, related to the diluent of the test solution. Samples of 1 mg/mL thiotepa infusion solutions were assayed without further dilution.

### 2.3. HPLC Assay

Concentrations of thiotepa were determined using a stability indicating reversed-phase high-performance liquid chromatography (RP-HPLC) method. Chromatographic conditions used were adapted from the thiotepa BP monograph and are given in Table 1. Degradation products were determined semi-quantitatively.

The RP-HPLC assay was carried out using a Waters Alliance 2695 pump, connected to a Waters photodiode array detector 2990 (Waters, Eschborn, Germany). Data were collected and processed using Waters Empower 2 Software, Version 6.10.01.00. Each sample was injected in triplicate by an autosampler. Peak areas of secondary peaks ≥ 0.1% of the main peak were calculated as a percentage of the main peak area [11].

### 2.4. Validation of the RP-HPLC Assay

The system suitability test (SST) was performed as described in the BP Thiotepa monographs by solving 10 mg thiotepa USP CRS in 2 mL methanol, adding 50 µL of 0.1% phosphoric acid and heating the sample at 65 °C for one minute to generate methoxy-thiotepa [10,12]. The RP-HPLC method was validated corresponding to the ICH Guideline Q2 (R1) Validation of analytical procedures [14]. The suitability of the method was tested by analysing forced degraded samples of thiotepa USP CRS dissolved in water HPLC grade to 1 mg/mL. Heat-degrading conditions were 80 °C for 60 min. Acidic and alkaline degradation were performed by adding 0.1 M hydrochloric acid, and 0.1 M sodium hydroxide, respectively. The samples were heated for 60 min at 80 °C and assayed after neutralization. The chloro-adduct impurity was identified by dissolving 15 mg thiotepa CRS in 10 mL water HPLC grade and adding 1 g sodium chloride. The sample was heated for 10 min at 100 °C [10]. 

Linearity of the method was tested with a stock solution of 20 mg thiotepa USP CRS dissolved in 10 mL water HPLC grade. By further dilution with G5 or 0.9% NaCl, solutions of the concentration 0.5 mg/mL, 0.8 mg/mL, 0.9 mg/mL, 1.0 mg/mL, 1.1 mg/mL, 1.2 mg/mL and 1.5 mg/mL were prepared as calibration standards and assayed (3 injections each). The calibration curve was attained by plotting the peak area versus the nominal thiotepa concentrations.

Intra- and inter-day accuracy and precision were tested by assaying ten thiotepa solutions on five consecutive days. A thiotepa 3 mg/mL stock solution was freshly prepared each day by solving 30 mg thiotepa USP CRS ad 10 mL with water HPLC grade. A total of 10 thiotepa test solutions of the nominal concentration 1 mg/mL were produced by mixing 0.33 mL stock solution with 0.67 mL G5. The first and tenth solution were injected tenfold, solution 2–9 onefold on day 1–5.

### 2.5. pH Measurement

The pH values were determined using a SevenCompact S210 pH meter equipped with an InLab Micro Pro-ISM electrode (Mettler Toledo, Greifensee, Switzerland). The instrument was calibrated weekly with standard buffer solutions of pH 2.00 (batch: 1F317E), pH 4.01 (batch: 1G207E), pH 7.00 (batch: 1G209C), pH 9.21 (batch: 1F316C), pH 11.00 (batch: 1F325F, Mettler Toledo, Greifensee, Switzerland). Functional testing was carried out on each measuring day with a 7.00 standard buffer solution (batch: 1G209C, accepted deviation: ±0.05). Samples withdrawn were measured once without further dilution.

### 2.6. Osmolality

The osmolality of samples withdrawn was measured once using an Osmomat 3000 D (Gonotec GmbH, Berlin, Germany). A two-point calibration was carried out weekly with a 300 mOsmol/kg calibration standard (Gonotec GmbH, Berlin, Germany) and water HPLC grade. Functional testing was carried out on each measuring day with water HPLC grade.

### 2.7. Determination of Visible and Sub-Visible Particles

All test solutions were inspected visually for visible particles and colour changes whenever samples were withdrawn. Sub-visible particles of the infusion solutions were determined directly after preparation, and on day 14 with a PAMAS SVSS particle counter equipped with an HCB-LD-50/50 sensor (PAMAS, Rutesheim, Germany) using USP V.3.6.3 software. The aspiration tube of the particle counter was inserted directly into the infusion bags with an attached cannula. According to Ph. Eur., 2.9.19, a volume of 5 mL test solution was withdrawn and moved through the laser-diode sensor automatically. Each test solution was measured four times; the first result was disregarded. According to Ph. Eur., 2.9.19 and USP <788> for parenteral infusion solutions in containers with a nominal volume of 100 mL or less, the number of 6000 particles ≥10 µm and 600 particles ≥25 µm is accepted.

## 3. Results

### 3.1. Validation of the RP-HPLC Assay

#### 3.1.1. Suitability

The SST chromatogram showed the thiotepa parent peak with a retention time (Rt) of 11 min and the methoxy-thiotepa peak with Rt 15 min and relative retention time (rRt) 1.36. The resolution factor of 7 between these two peaks and the symmetry factor of 0.9 for thiotepa met the pharmacopoeia requirements relevant at the time of the investigation (>3 [10]; 0.8–1.5 [15]).

All chromatograms of forced degraded samples (see Figure 2) showed several peaks of hydrophilic degradation products with rRts < 0.27 compared to the parent peak, partly overlaid by the injection peak. Under acidic and heat conditions, the thiotepa parent peak was no longer detectable, whereas under alkaline and heat conditions, the thiotepa peak area was distinctly reduced. Heat-degraded solutions revealed two secondary peaks with Rt 6 min (rRt 0.44) and 10 min (rRt 0.77). No secondary peak interfered with the thiotepa parent peak. The chloro-adduct assay showed a peak corresponding to the monochloro-adduct at Rt 45 min (rRt 3.6) in line with the pharmacopoeia monograph specifications (rRt about 3.75) [12,13].

#### 3.1.2. Linearity and Intra-/Inter-Day Precision

The coefficient of correlation attained by plotting the peak areas against the nominal thiotepa concentrations was R^2^ = 0.999, thereby proving linearity of the assay. The intra-day precision assay revealed a mean thiotepa concentration of 0.996 mg/mL (99.6%) ± 0.43% relative standard deviation (RSD). The inter-day precision assay revealed a mean thiotepa concentration of 0.991 mg/mL (99.1%) ± 0.33% RSD. The results meet the criteria based on ICH Q2 (R1) and proved reproducibility.

### 3.2. Thiotepa Concentrate 10 mg/mL

Detailed results of the stability tests of thiotepa concentrate are shown in Table 2. When stored refrigerated for 14 days, no decrease in thiotepa concentration was determined and no secondary peaks became visible. However, when test solutions were stored at 25 °C, thiotepa concentrations fell below the 95% limit already after day 3 of storage. From day 7 onward, up to three small secondary peaks in the rRt range > 0.27 and <0.8 became obvious in the related HPLC-chromatograms (compare Figure 3). The percentage rates of the peak areas of secondary peak no. 2, 3, 4 are given in Table 2. Secondary peaks eluting with rRts < 0.27 were not quantitatively analysed. 

The pH values of the concentrates amounted to pH 7 immediately after reconstitution and slowly increased to pH 7.5 (upper limit) on day 14 when kept refrigerated. When stored at 25 °C, the pH increased more rapidly, exceeded the pH 7.5 limit after one day of storage and ended at pH 8.2 on day 14. 

White particles were visible in the concentrates stored at room temperature and under refrigeration from day 3 and day 5 onwards, respectively.

### 3.3. Thiotepa Infusion Solutions Diluted with Glucose 5%

Thiotepa concentrations remained above 95% of the initial concentrations for 14 days when test solutions were kept refrigerated, irrespective of the nominal concentration (1, 2, 3 mg/mL). When stored at 25 °C, thiotepa concentrations remained above 95% of the initial concentration until day 3 in 1 mg/mL test solutions, day 5 in 2 mg/mL test solutions, and day 7 in 3 mg/mL test solutions. Detailed results of the quantitative HPLC-analysis and pH values of thiotepa infusion solutions diluted with G5 are given in Table 3. Chromatograms of the thiotepa test solutions showed the three secondary peaks no.1, 2, and 3 eluting with the rRt 0.44, 0.50, and 0.52 (see Figure 4), mainly in solutions stored at room temperature, partially exceeding the 0.2% threshold for impurities. In addition, several peaks eluted at Rt 1–3.5 min and peak areas increased over the storage time. They were partly related to the glucose diluent solution and partly related to hydrophilic thiotepa degradation products. Correct attribution and semi-quantitative determination of these peaks were not possible.

The initial pH values were slightly acidic (about pH 6.2–6.5) regardless of the nominal concentrations. Over the 14-day observation period, the pH values increased in all test solutions. The increase of pH up to 7.5 was accelerated in higher concentrated solutions and at the higher storage temperature (compare Table 3). The osmolality of the infusion solutions was related to the nominal thiotepa concentrations (1 mg/mL = 266 mOsmol/kg, 2 mg/mL = 245 mOsmol/kg, 3 mg/mL = 218 mOsmol/kg) and remained unchanged over the 14-day observation period. In none of the solutions were visible particles detected. Sub-visible particles were in accordance with pharmacopoeia requirements over the entire observation period (7–74 particles ≥25 µm, 431–2736 particles ≥10 µm).

### 3.4. Thiotepa Infusion Solutions Diluted with 0.9% NaCl

All test solutions stored at 2–8 °C revealed a thiotepa concentration above 95% of the initial concentration over the 14 days storage period. When stored at 25 °C, thiotepa concentrations stayed above 95% until day 5 in 1 mg/mL solutions and day 7 in 2 mg/mL and 3 mg/mL test solutions (see Table 4).

The monochloro-adduct was not detected in any of the test solutions stored at 2–8 °C. Chromatograms of samples stored at 25 °C showed a peak referring to the monochloro-adduct (rRt 3.6), exceeding the chosen specification of 0.6% after day 5 (1 mg/mL) or day 7 (2 mg/mL, 3 mg/mL) (compare Figure 5). In addition, the secondary peak no. 1 (rRt 0.44), 3 (rRt 0.54) and 4 (rRt 0.77) became obvious. Secondary peak no. 3 and no. 4 exceeded the 0.2% threshold for impurities on day 14 (1 mg/mL, 2 mg/mL) or day 28 (3 mg/mL) (compare Table 4). Additional peaks eluted at the solvent front and peaks with rRt < 0.27 min were not determined quantitatively. Peak areas partly increased over time. 

In all test solutions, an increase of pH values from an average of 6.5 to a maximum of 8.3 was observed over time, particularly in higher concentrated solutions stored at 25 °C without light protection (Table 4). The osmolality of test solutions correlated inversely with the thiotepa concentration (1 mg/mL = 256 mOsmol/kg, 2 mg/mL = 235 mOsmol/kg, 3 mg/mL = 216 mOsmol/kg) and stayed unchanged over the whole study period in all test solutions. No visible particles or colour changes were observed in diluted infusion solutions. Sub-visible particles were in accordance with pharmacopoeia requirements over the entire observation period (8–233 particles ≥25 µm, 548–2593 particles ≥10 µm), except for thiotepa 3 mg/mL solutions on day 0. This finding is acknowledged as an outlier.

## 4. Discussion

The in-use stability tests of thiotepa concentrate and diluted thiotepa infusion solutions were performed according to the ‘Good Stability Testing’ guidelines [16,17]. Suitable RP-HPLC methods are described in the pharmacopoeia and further publications [5,6,10,11]. The stability-indicating nature of the chosen BP HPLC assay was clearly proven in the forced degradation experiments. The resulting chromatograms showed peaks of degradation products related to the absence/presence of chloride ions and pH in accordance with the results of known degradation-time profiles [7]. The mobile phase contained 15% acetonitrile eluting the thiotepa parent peak with a retention time of about 13 min. The prolonged Rt enabled detection and partial quantification of several peaks of hydrophilic degradation products with rRts < 1. However, it must be assumed that several hydrophilic degradation products are eluted in the eluting front and/or are not detectable by the UV detection method (e.g., aziridine). Standards for the thiotepa degradation products are not listed in the pharmacopoeial standard catalogues. For precise identification of the degradation peaks, further analysis, such as HPLC-MS, would be adequate. Further experimental identification was not performed because degradation pathways and degradation products are already published in the scientific literature. It should be noted that we focussed on the physicochemical stability of highly concentrated ready-to-administer thiotepa infusion solutions relevant for daily clinical practice in advanced oncology centres. Concentrations of the diluted test solutions, type of diluent infusion solutions (0.9% NaCl, G5), primary containers (polyolefin bags), and storage conditions (2–8 °C, 25 °C) were chosen with an eye to clinical practice. As thiotepa is not considered photosensitive [3], worst-case lightening over 24 h was applied in the climatic chamber. The stability limit set for the remaining thiotepa concentration was 95% (instead of ≥90%) of the nominal concentration because of the known instability of reconstituted thiotepa solutions in combination with the precautionary principle. 

The degradation rates and degradation pattern of the investigated thiotepa test solutions were clearly dependent on temperature, pH, thiotepa concentration and presence of chloride ions. The observed elevation of pH values is consistent with the thiotepa degradation mechanism especially pseudo-first order hydrolysis reactions and chloride-adduct formation, thereby yielding primary amines. In total, four secondary peaks with rRts ≥ 0.27 and <1 were quantitatively analysed, most probably attributable to hydroxythiotepa (impurity A mentioned in the BP [12]) and intermediate partly hydroxylated and chlorinated thiotepa degradation products (among others, monohydroxyethylamine-adduct = impurity B mentioned in the BP [12]). Because these peaks are related to intermediate degradation products, the peak areas did not increase significantly over time but varied erratically at different sampling points. In chloride-containing test solutions stored at 25 °C, the peak areas of monochloro-adduct increased over time and exceeded the specification limit after prolonged storage. Time-dependent increase in the monochloro-adduct concentration and decrease in the thiotepa concentration became particularly clear when the analysis of 3 mg/mL thiotepa test solutions stored at 25 °C was postponed to day 28 of storage (instead of day 14) due to technical problems. Unchanged osmolality over the whole observation period could be interpreted as an inadequate degradation parameter.

Information about the stability of reconstituted thiotepa concentrates is limited. According to the EPAR, suitable stability was demonstrated for 24 h at 2–8 °C and for 12 h at 25 °C [3]. In our experiments, the potency of reconstituted thiotepa concentrate 10 mg/mL remained unchanged over the 14-day observation period, when stored refrigerated. In addition, the pH of the refrigerated concentrates remained in the specification limits. Storage at a higher temperature caused significant degradation, accompanied by pH elevation as well as particle formation, and should be avoided. Visible particles, resulting from polymerization of thiotepa, became obvious in the thiotepa concentrates stored in the original glass vials regardless of the storage temperature. As particle formation did not affect the potency of the concentrates stored under refrigeration, in-line filtration allows further use of refrigerated solutions (compare SmPC). In accordance with our results, Thiotepa Riemser concentrate was reported very recently to be stable for 28 days when stored at 2–8 °C [18]. According to the SmPC, the powder for concentrate for infusion solution is also to be stored and transported refrigerated [2]. 

Storage temperature is also the most relevant stability-determining factor for diluted thiotepa solutions. All test solutions kept refrigerated met the specification of ≥95% of the initial thiotepa concentration over the 14-day observation period, whereas storage at 25 °C led to a substantial decrease of thiotepa concentrations after day 3, correlated to the nominal concentration. The lower concentrated 1 mg/mL thiotepa solutions are more prone to degradation reactions than higher concentrated solutions (2 mg/mL, 3 mg/mL). Lyophilised thiotepa, diluted in G5 infusion solution to the concentrations 0.5 mg/mL and 5 mg/mL, is reported to remain stable for 8 h and 14 days at 4 °C, respectively [5]. Storage at 23 °C resulted in 8-h and 3-day stability [5]. Concentration-dependent degradation rates were also found when initial thiotepa concentrations amounted to 0.5 mg/mL, 1 mg/mL and 3 mg/mL, and 0.9% NaCl infusion solution was used as diluent [6]. Over the observation period of 48 h, the decrease in the thiotepa concentration was greater than in our experiments. This might be explained by the different thiotepa brand products investigated and the prefilled infusion bags used. In general, the pH value of 0.9% NaCl infusion solution is lower in prefilled PVC bags (pH 5.0 [19]) than in the prefilled polyolefin bags used in our experiments (pH 6.2). More acidic conditions significantly accelerate the degradation reactions of the protonated activated aziridinium moieties. The short in-use stability of 0.5 mg/mL to 1 mg/mL Thiotepa Riemser in 0.9% NaCl infusion given in the SmPC is most probably related to the unfavourable low thiotepa concentrations and the chloride-containing diluent recommended [2]. In addition, differing specifications set by the manufacturer, such as the monochloro-adduct concentration, may explain the reduced in-use stability given in the SmPC.

Chloro-adducts are a result of the chemical reaction between thiotepa and chloride ions. The amount and rate of monochloro-adduct formation is inversely related to the thiotepa concentration [6]. In the BP monograph, the upper impurity limit of the monochloro-adduct is defined as 0.15% of the thiotepa peak area. It should be noted that formerly, the FDA accepted thiotepa solutions containing less than 0.6% chloro-adduct as stable and acceptable for use [6]. In chloride-containing thiotepa solutions stored at room temperature, the excessive formation of the chloro-adduct occurs inevitably. The chloro-adduct is also formed in biological samples such as plasma and urine, where the half-life of thiotepa depends on the pH and temperature (most rapid degradation in acidic urine at 37 °C) [20]. Probably the chloro-adduct is also formed as an intermediate metabolite in patients. The genotoxicity of the thiotepa monochloro-adduct is related to the alkylating properties of the remaining aziridine moieties and is most probably not significantly toxicologically different to thiotepa [21]. Therefore, the loss of potency is assumed as the most relevant factor for the determination of in-use stability of thiotepa and 0.6% chloro-adduct as specification accepted. The toxicity of partially hydroxylated degradation products (like secondary peak no. 1 to 4) is assessed likewise and indicated as the threshold related to the BP specification for impurities. 

However, the formation of the chloro-adduct can easily be avoided by using G5 as a diluent solution. In our stability studies, the initial pH values of thiotepa test solutions prepared in prefilled G5 infusion bags were equivalent to test solutions in prefilled 0.9% NaCl polyolefin bags. pH values of different types of prefilled G5 infusion bags are nearly identical (PVC pH 4.1 [19], polyolefin pH 4.5). Consequently, the stability studies of Xu et al. showed a similar loss of thiotepa regardless of the type of container [5]. The stability reported for the 5 mg/mL test solutions was comparable to our results related to the storage temperature and period. Compared to 0.9% NaCl as a diluent, the degradation rate is reduced correlated to a less rapid increase in the pH. To avoid increased degradation rates, nominal thiotepa concentrations of diluted solutions should be ≥1 mg/mL. The use of G5 as diluent solution is favourable, but is not named in the official documentation of Thiotepa Riemser. With regard to in-use stability for conditioning therapies, 2 mg/mL to 3 or 5 mg/mL thiotepa infusion solutions in prefilled G5 infusion bags is preferred.

## 5. Conclusions

The physicochemical stability of thiotepa test solutions is taken for granted when the specifications set (≥95% of the initial thiotepa concentration, pH-value of the concentrate 5.5–7.5, monochloro-adduct ≤ 0.6% of the thiotepa parent peak area, and pharmacopoeia specification for particle counts) are fulfilled at the same time.

Thiotepa concentrates are stable for at least 14 days stored at 2–8 °C, or 24 h stored at 25 °C. Thiotepa 1 mg/mL, 2 mg/mL, 3 mg/mL infusion solutions in G5 are stable for at least 14 days stored at 2–8 °C, or 3 days (1 mg/mL), 5 days (2 mg/mL) or 7 days (3 mg/mL) stored at 25 °C. Thiotepa 1 mg/mL, 2 mg/mL, 3 mg/mL infusion solutions in 0.9% NaCl are stable for at least 14 days stored at 2–8 °C, or 5 days (1 mg/mL) or 7 days (2 mg/mL, 3 mg/mL) stored at 25 °C.

Due to the temperature-dependent physicochemical stability and for microbiological reasons, thiotepa concentrates and infusion solutions should be stored consistently at 2–8 °C. Because of the increased formation of chloro-adducts in thiotepa infusion solutions diluted with 0.9% NaCl, the use of infusion bags prefilled with glucose 5% is recommended as a vehicle solution. Because of the inversely correlated degradation rates, nominal thiotepa concentrations should be higher than 1 mg/mL. 

## Figures and Tables

**Figure 2 pharmaceutics-15-00309-f002:**
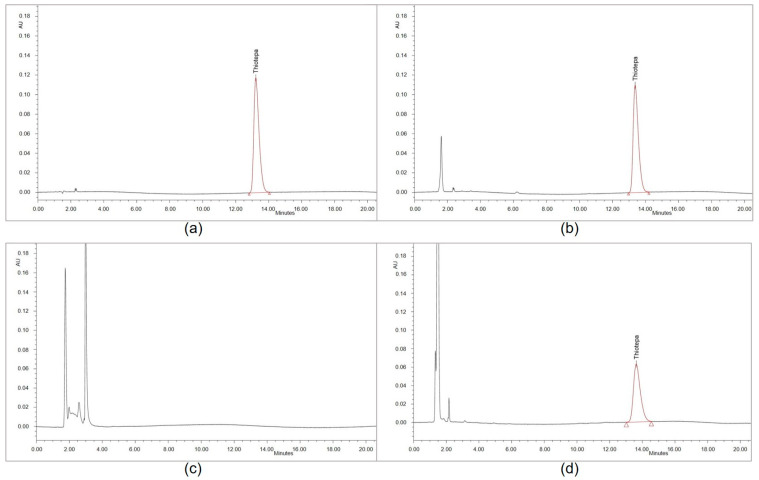
HPLC chromatograms of thiotepa in water without degradation (**a**), heat-degraded (**b**), acidic and heat-degraded (**c**) alkaline and heat-degraded (**d**).

**Figure 3 pharmaceutics-15-00309-f003:**
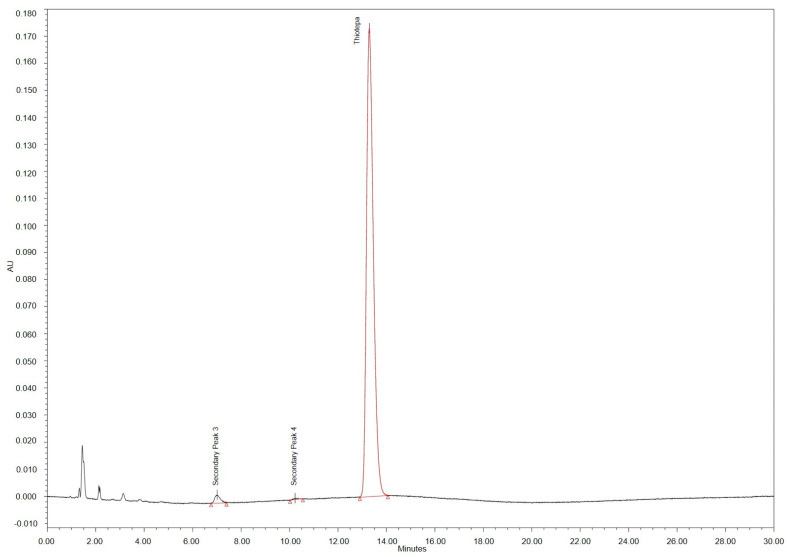
HPLC chromatogram of thiotepa 10 mg/mL concentrate after 14 days storage at 25 °C.

**Figure 4 pharmaceutics-15-00309-f004:**
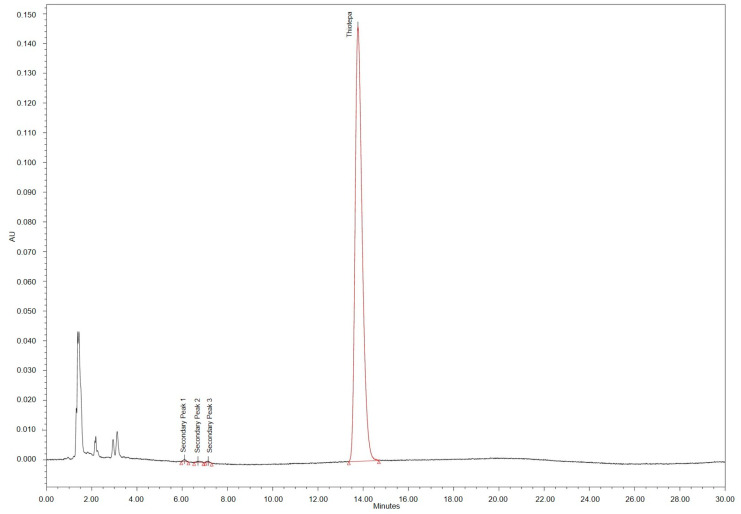
HPLC chromatogram of thiotepa 1 mg/mL infusion diluted with G5 infusion solution and stored at 25 °C for 14 days.

**Figure 5 pharmaceutics-15-00309-f005:**
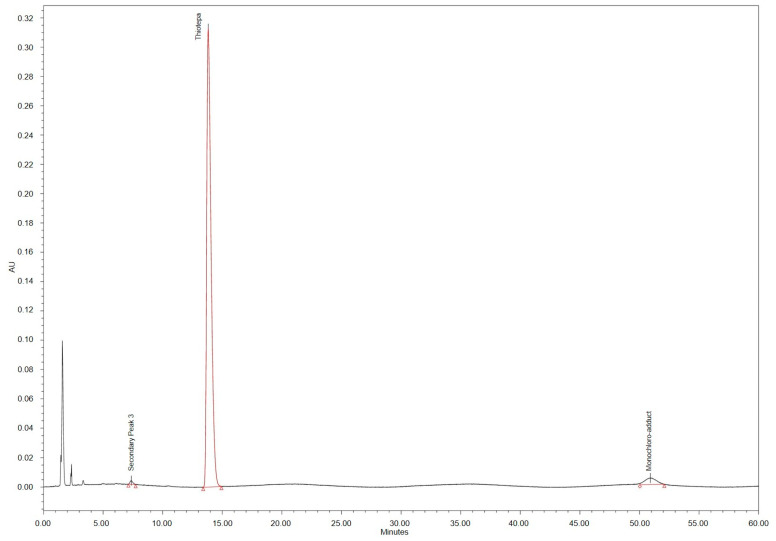
HPLC chromatogram of thiotepa 1 mg/mL infusion diluted with 0.9% NaCl solution stored at 25 °C for 14 days.

**Table 1 pharmaceutics-15-00309-t001:** Parameters of the RP-HPLC assay.

Chromatography Parameter	Condition/Set Value
Column	Nucleodur 100 C_18_, 5 µm, 150 × 4.6 mm (Macherey-Nagel)
Flow rate	1.0 mL/min
Injection volume	10 µL of thiotepa concentrate, thiotepa infusion solutions in G5 20 µL of thiotepa infusion solutions in 0.9% NaCl
Sample temperature	10 °C
Column temperature	30 °C
Run time	30 min for thiotepa concentrate, thiotepa infusion solutions in G5 60 min for thiotepa infusion solutions in 0.9% NaCl, to detect the monochloro-adduct
Detection wavelength	215 nm
Mobile phase	15% acetonitrile and 85% 0.1 M phosphate buffer pH 7
Pump mode	Isocratic

**Table 2 pharmaceutics-15-00309-t002:** Thiotepa concentration, peak areas of secondary peaks and pH of thiotepa concentrate 10 mg/mL in the punctured vial stored under two different conditions for 14 days.

Storage Conditions	Initial Thiotepa Concentration			Percentage Rate of the Initial Thiotepa Concentration ± RSD [%] (Concentration at d 0 = 100%)									
	[mg/mL] ± RSD [%]			Peak Area of Impurities [%] in Relation to Thiotepa Peak Area									
				n = 9									
	Nominal	Measured (d 0)		d 1		d 3		d 5		d 7		d 14	
2–8 °C light protected	10 mg/mL	thiotepa	10.05 ± 0.88	thiotepa	100.11 ± 0.38	thiotepa	100.29 ± 0.84	thiotepa	97.91 * ± 2.18	thiotepa	101.01 * ± 0.46	thiotepa	101.06 * ± 0.71
25 °C in daylight	10 mg/mL	thiotepa	10.19 ± 0.83	thiotepa	98.23 ± 0.64	thiotepa	95.96 * ± 0.62	thiotepa	93.06 * ± 2.65	thiotepa	93.67 * ± 1.16	thiotepa	85.89 * ± 0.35
		sec. peak 2	n.d.	sec. peak 2	n.d.	sec. peak 2	n.d.	sec. peak 2	n.d.	sec. peak 2	≥0.2% (0.33%)	sec. peak 2	n.d.
		sec. peak 3	n.d.	sec. peak 3	n.d.	sec. peak 3	n.d.	sec. peak 3	n.d.	sec. peak 3	n.d.	sec. peak 3	≥0.2% (1.48%)
		sec. peak 4	n.d.	sec. peak 4	n.d.	sec. peak 4	n.d.	sec. peak 4	n.d.	sec. peak 4	≤0.2% (0.15%)	sec. peak 4	≤0.2% (0.14%)
		pH-values of thiotepa											
2–8 °C light protected	10 mg/mL	7.1 ± 0.7		6.9 ± 3.8		7.0 ± 1.5		7.0 ± 2.2		7.3 ± 1.6		7.5 ± 0.8	
25 °C in daylight	10 mg/mL	7.0 ± 0.7		7.4 ± 1.3		7.7 ± 0.3		7.9 ± 0.1		8.0 ± 0.2		8.2 ± 0.4	

* particles were visible in the test solutions at this point in time. n.d. = not detected.

**Table 3 pharmaceutics-15-00309-t003:** Thiotepa concentration, peak areas of secondary peaks and pH of thiotepa infusion solutions (1 mg/mL, 2 mg/mL, 3 mg/mL) diluted in glucose 5% carrier solution stored under two different conditions for 14 days.

Storage Conditions	Initial Thiotepa Concentration			Percentage Rate of the Initial Thiotepa Concentration ± RSD [%] (Concentration at d 0 = 100%)									
	[mg/mL] ± RSD [%]			Peak Area of Impurities [%] in Relation to Thiotepa Peak Area									
				n = 9									
	Nominal	Measured (d 0)		d 1		d 3		d 5		d 7		d 14	
2–8 °C light protected	1 mg/mL	thiotepa	0.95 ± 1.08	thiotepa	99.03 ± 1.01	thiotepa	100.20 ± 1.16	thiotepa	99.14 ± 1.11	thiotepa	99.38 ± 0.89	thiotepa	98.37 ± 1.06
		sec. peak 1	n.d.	sec. peak 1	n.d.	sec. peak 1	n.d.	sec. peak 1	≤0.2% (0.14%)	sec. peak 1	n.d.	sec. peak 1	n.d.
	2 mg/mL	thiotepa	1.94 ± 0.79	thiotepa	100.31 ± 0.83	thiotepa	100.02 ± 0.81	thiotepa	99.25 ± 0.78	thiotepa	100.13 ± 0.77	thiotepa	99.31 ± 0.89
		sec. peak 1	≤0.2% (0.17%)	sec. peak 1	≤0.2% (0.14%)	sec. peak 1	≤0.2% (0.12%)	sec. peak 1	≤0.2% (0.14%)	sec. peak 1	≤0.2% (0.14%)	sec. peak 1	≤0.2% (0.14%)
	3 mg/mL	thiotepa	2.85 ± 1.96	thiotepa	99.68 ± 2.09	thiotepa	100.61 ± 1.98	thiotepa	100.36 ± 2.17	thiotepa	100.62 ± 1.97	thiotepa	99.51 ± 2.14
		sec. peak 1	≤0.2% (0.15%)	sec. peak 1	≤0.2% (0.12%)	sec. peak 1	≤0.2% (0.14%)	sec. peak 1	≤0.2% (0.15%)	sec. peak 1	≤0.2% (0.15%)	sec. peak 1	≤0.2% (0.11%)
25 °C in daylight	1 mg/mL	thiotepa	0.94 ± 0.94	thiotepa	97.53 ± 1.13	thiotepa	97.14 ± 1.08	thiotepa	94.54 ± 1.18	thiotepa	93.30 ± 1.29	thiotepa	87.74 ± 1.29
		sec. peak 1	≤0.2% (0.14%)	sec. peak 1	≤0.2% (0.19%)	sec. peak 1	≥0.2% (0.29%)	sec. peak 1	≥0.2% (0.32%)	sec. peak 1	≥0.2% (0.36%)	sec. peak 1	≥0.2% (0.25%)
		sec. peak 2	n.d.	sec. peak 2	n.d.	sec. peak 2	n.d.	sec. peak 2	n.d.	sec. peak 2	≤0.2% (0.12%)	sec. peak 2	≤0.2% (0.19%)
		sec. peak 3	n.d.	sec. peak 3	n.d.	sec. peak 3	n.d.	sec. peak 3	n.d.	sec. peak 3	n.d.	sec. peak 3	≤0.2% (0.12%)
	2 mg/mL	thiotepa	1.92 ± 0.53	thiotepa	99.43 ± 0.21	thiotepa	97.27 ± 0.54	thiotepa	95.03 ± 0.62	thiotepa	94.56 ± 0.38	thiotepa	90.87 ± 2.98
		sec. peak 1	≤0.2% (0.15%)	sec. peak 1	≥0.2% (0.28%)	sec. peak 1	≥0.2% (0.29%)	sec. peak 1	≥0.2% (0.30%)	sec. peak 1	≥0.2% (0.27%)	sec. peak 1	≥0.2% (0.24%)
		sec. peak 3	n.d.	sec. peak 3	n.d.	sec. peak 3	n.d.	sec. peak 3	n.d.	sec. peak 3	n.d.	sec. peak 3	≥0.2% (0.65%)
	3 mg/mL	thiotepa	2.81 ± 2.86	thiotepa	99.67 ± 2.91	thiotepa	97.93 ± 2.74	thiotepa	97.03 ± 2.87	thiotepa	95.04 ± 2.88	thiotepa	91.33 ± 3.02
		sec. peak 1	≤0.2% (0.13%)	sec. peak 1	≤0.2% (0.18%)	sec. peak 1	≤0.2% (0.2%)	sec. peak 1	≥0.2% (0.36%)	sec. peak 1	≥0.2% (0.37%)	sec. peak 1	≤0.2% (0.16%)
		sec. peak 2	n.d.	sec. peak 2	n.d.	sec. peak 2	n.d.	sec. peak 2	n.d.	sec. peak 2	n.d.	sec. peak 2	≥0.2% (0.85%)
		pH-values of thiotepa											
2–8 °C light protected	1 mg/mL	6.7 ± 4.8		7.0 ± 1.9		6.9 ± 2.5		6.8 ± 1.7		7.1 ± 2.0		7.3 ± 1.3	
	2 mg/mL	6.5 ± 4.2		6.9 ± 4.9		7.0 ± 2.9		7.4 ± 2.8		7.4 ± 1.1		7.0 ± 2.9	
	3 mg/mL	6.3 ± 0.3		7.2 ± 0.2		7.1 ± 1.2		7.1 ± 0.5		7.1 ± 1.4		7.1 ± 0.4	
25 °C in daylight	1 mg/mL	6.4 ± 2.0		7.2 ± 0.2		7.1 ± 0.4		7.1 ± 0.2		7.1 ± 0.1		7.2 ± 0.2	
	2 mg/mL	6.5 ± 2.8		7.0 ± 0.9		7.1 ± 1.7		7.3 ± 0.2		7.4 ± 0.5		7.5 ± 0.4	
	3 mg/mL	6.2 ± 0.5		7.2 ± 0.8		7.3 ± 0.5		7.4 ± 0.5		7.5 ± 0.4		7.6 ± 0.2	

n.d. = not detected.

**Table 4 pharmaceutics-15-00309-t004:** Thiotepa concentration, peak areas of secondary peaks and pH of thiotepa infusion solutions (1 mg/mL, 2 mg/mL, 3 mg/mL) diluted in 0.9% NaCl carrier solution stored under two different conditions for 14 days.

Storage Conditions	Initial Thiotepa Concentration			Percentage Rate of the Initial Thiotepa Concentration ± RSD [%] (Concentration at d 0 = 100%)									
	[mg/mL] ± RSD [%]			Peak Area of Impurities [%] in Relation to Thiotepa Peak Area									
				n = 9									
	Nominal	Measured (d 0)		d 1		d 3		d 5		d 7		d 14	
2–8 °C light protected	1 mg/mL	thiotepa	0.98 ± 2.04	thiotepa	99.10 ± 2.13	thiotepa	98.60 ± 2.15	thiotepa	98.25 ± 2.02	thiotepa	97.38 ± 2.25	thiotepa	96.68 ± 2.11
		monochloro-adduct	n.d.	monochloro-adduct	n.d.	monochloro-adduct	n.d.	monochloro-adduct	n.d.	monochloro-adduct	n.d.	monochloro-adduct	n.d.
	2 mg/mL	thiotepa	1.92 ± 0.24	thiotepa	99.01 ± 0.30	thiotepa	99.01 ± 0.75	thiotepa	98.86 ± 0.37	thiotepa	99.21 ± 0.64	thiotepa	97.54 ± 0.67
		sec. peak 1	n.d.	sec. peak 1	≤0.2% (0.10%)	sec. peak 1	≤0.2% (0.10%)	sec. peak 1	n.d.	sec. peak 1	n.d.	sec. peak 1	n.d.
		monochloro-adduct	n.d.	monochloro-adduct	n.d.	monochloro-adduct	n.d.	monochloro-adduct	n.d.	monochloro-adduct	n.d.	monochloro-adduct	n.d.
	3 mg/mL	thiotepa	2.90 ± 1.08	thiotepa	100.22 ± 0.80	thiotepa	99.87 ± 0.75	thiotepa	100.69 ± 1.58	thiotepa	99.59 ± 1.71	thiotepa	98.86 ± 0.88
		monochloro-adduct	n.d.	monochloro-adduct	n.d.	monochloro-adduct	n.d.	monochloro-adduct	n.d.	monochloro-adduct	n.d.	monochloro-adduct	n.d.
25 °Cin daylight	1 mg/mL	thiotepa	0.97 ± 1.98	thiotepa	98.33 ± 2.05	thiotepa	98.12 ± 1.55	thiotepa	95.48 ± 1.78	thiotepa	94.09 ± 1.96	thiotepa	91.40 ± 4.18
		sec. peak 1	≤0.2% (0.10%)	sec. peak 1	n.d.	sec. peak 1	n.d.	sec. peak 1	n.d.	sec. peak 1	n.d.	sec. peak 1	n.d.
		sec. peak 3	n.d.	sec. peak 3	n.d.	sec. peak 3	n.d.	sec. peak 3	n.d.	sec. peak 3	n.d.	sec. peak 3	≥0.2% (0,59%)
		monochloro-adduct	n.d.	monochloro-adduct	n.d.	monochloro-adduct	≤0.6% (0.10%)	monochloro-adduct	≤0.6% (0.6%)	monochloro-adduct	≥ 0.6% (1.58%)	monochloro-adduct	≥0.6% (3.46%)
	2 mg/mL	thiotepa	1.92 ± 0.73	thiotepa	98.69 ± 0.47	thiotepa	97.58 ± 0.85	thiotepa	96.57 ± 0.48	thiotepa	95.51 ± 0.57	thiotepa	88.69 ± 1.06
		sec. peak 1	n.d.	sec. peak 1	≤0.2% (0.10%)	sec. peak 1	n.d.	sec. peak 1	n.d.	sec. peak 1	n.d.	sec. peak 1	n.d.
		sec. peak 3	n.d.	sec. peak 3	n.d.	sec. peak 3	n.d.	sec. peak 3	n.d.	sec. peak 3	n.d.	sec. peak 3	≥0.2% (0,26%)
		monochloro-adduct	n.d.	monochloro-adduct	n.d.	monochloro-adduct	n.d.	monochloro-adduct	n.d.	monochloro-adduct	≤0.6% (0.47%)	monochloro-adduct	≥0.6% (0.87%)
	3 mg/mL	thiotepa	2.84 ± 0.23	thiotepa	99.64 ± 0.30	thiotepa	97.90 ± 0.21	thiotepa	98.12 ± 1.32	thiotepa	95.20 ± 0.12	thiotepa	d 28 * 76.98 ± 0.63
		sec. peak 3	n.d.	sec. peak 3	n.d.	sec. peak 3	n.d.	sec. peak 3	n.d.	sec. peak 3	n.d.	sec. peak 3	≥0.2% (2.94%)
		sec. peak 4	n.d.	sec. peak 4	n.d.	sec. peak 4	n.d.	sec. peak 4	n.d.	sec. peak 4	n.d.	sec. peak 4	≥0.2% (0.74%)
		monochloro-adduct	n.d.	monochloro-adduct	n.d.	monochloro-adduct	n.d.	monochloro-adduct	≤0.6% (0.18%)	monochloro-adduct	≤0.6% (0.34%)	monochloro-adduct	**≥0.6% (3.77%)**
		pH-values of thiotepa											
2–8 °C light protected	1 mg/mL	6.8 ± 3.2		7.0 ± 2.3		6.8 ± 3.5		6.8 ± 2.2		6.8 ± 2.0		7.0 ± 1.6	
	2 mg/mL	6.7 ± 1.9		6.9 ± 1.3		6.9 ± 2.0		6.5 ± 2.3		6.8 ± 2.3		7.0 ± 0.7	
	3 mg/mL	6.4 ± 1.4		6.8 ± 0.7		6.6 ± 1.7		6.7 ± 0.3		6.9 ± 0.4		7.2 ± 0.3	
25 °C in daylight	1 mg/mL	6.3 ± 1.2		6.9 ± 1.3		7.4 ± 1.0		7.5 ± 0.6		7.7 ± 0.3		7.9 ± 0.1	
	2 mg/mL	6.4 ± 1.1		7.1 ± 0.8		7.7 ± 3.1		7.9 ± 0.9		8.0 ± 0.1		8.2 ± 0.2	
	3 mg/mL	6.4 ± 0.6		6.8 ± 1.3		7.8 ± 1.0		8.1 ± 0.4		8.1 ± 0.3		8.3 ± 0.2	

* Due to a device failure, measurements planned for day 14 had to be postponed to day 28. n.d. = not detected.

## Data Availability

Not applicable.

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
