# Peer review of "Physicochemical Stability of Generic Thiotepa Concentrate and Ready-to-Administer Infusion Solutions for Conditioning Treatment"

_pharmaceutics, 2023, doi:10.3390/pharmaceutics15020309_

Round 1

Reviewer 1 Report

The researches presented results concerning the forced degradation / stability testing of Thiotepa infusion solutions when stored under refrigerated conditions and room temperature.

Manuscript is very simple and useful for hospital employees, pharmacists of hospital pharmacies. There are presented results obtained from simple analytical methods, typically used and recommended by pharmacopoeias. Therefore, degradation compounds which are detected cannot be identified. From the scientists point of view this manuscript does not present any problem solution or scientific hypothesis explanation. I can see none scientific novelty.

 Besides, I have some comments:

1.      Abstract, line 18 write glucose 5% instead of G5.

2.      I’ve noticed that photostability testing might be interesting . In my opinion there should be added results obtained for samples stored 24h but protected from light. ICH Q1B recommends using Suntest CPS+.

3.      Table 1 presents the same HPLC conditions as BP monograph, so it is not crucial to present it in this paper.

4.      Paragraph 2.4, line 164, I don’t understand the purpose of additional heating sample for 10 minutes at 100 °C, can you explain?

5.      Line 215, please add “under acidic and heat conditions” line 216 the same: “under alkaline and heat conditions”

6.      Line 219: detected peak corresponding to the monochloro-adduct at Rt 45 min may be presented on the chromatogram as Supplementary material.

7.      Section 3.2. consider additional results at 25 °C at dark conditions (aluminium foil).

8.      Table 2-Table 4 should be combine and presented in whole page. It would be clearer and the font will be bigger and more readable.

9.      Please, consider combining the Figure3.-Figure5. (separated a,b,c as in Fig.2 or overlayered).

10.  That’s pity that authors didn’t use any degradation compounds standards or pharmacopoeia impurity standards during analysis. Additionally, they didn’t apply the MS/MS detector to analyze degradants. Currently, the most important recommendation in pharmaceutical industry is to identify degradants and estimate ists genotoxic risk. Discussion based on guesses and probabilities is not valuable for science.

11.  It’s worth to prepare a comparison table from Discussion section (lines 267-402) including previous results and currently obtained by authors.

12.  Reference [15] – use the website https://www.edqm.eu/en/-/general-chapter-2.2.46.-chromatographic-separation-techniques-now-published-in-ph.-eur.-11th-edition

13.  Reference 16 is not available.

14.  Reference [17] use a website

15.  Reference [18] is not available – use the website https://www.krankenhauspharmazie.de/heftarchiv/2022/02/physikalisch-chemische-stabilitaet-thiotepa-haltiger-konzentrierter-loesungen.html

Reviewer 2 Report

The analysis of the physicochemical stability of some Thiotepa preparations as a function of temperature and time is reported. The analysis was performed using the reversed-phase HPLC analytical method. In general terms, the stability of Thiotepa depends on temperature and concentration.

Some comments on the manuscript are presented below. 

1. Title: The number of words in the title can be reduced. 

2. Abstract: Present a short context of the research. 

3. Introduction: The authors provide an excellent contextualization of the subject, the knowledge gap is well established and the objective of the paper is presented at the end. 

4. Materials and Methods: The materials used in the research are well characterized. The methods used are well specified, which allows the quality of the results to be verified and could be validated by other researchers.

5. Results: The results of the research show the experimental quality, allow to clearly and sufficiently demonstrate the fulfillment of the research objective. The structure of the tables and images is orderly and facilitates the reading and understanding of the manuscript. 

6. Except for the lack of discussion of the osmolarity results, the discussion of the results is clear and forceful. 

7. Conclusions and References: OK

Round 2

Reviewer 1 Report

Thank you for your reply. I accept your answers and please try to prepare nice readable tables..